# Investigating Causal Associations between the Gut Microbiota and Dementia: A Mendelian Randomization Study

**DOI:** 10.3390/nu16193312

**Published:** 2024-09-30

**Authors:** Zhi-Yuan Xiong, Hong-Min Li, Cheng-Shen Qiu, Xu-Lian Tang, Dan-Qing Liao, Li-Ying Du, Shu-Min Lai, Hong-Xuan Huang, Bing-Yun Zhang, Ling Kuang, Zhi-Hao Li

**Affiliations:** Department of Epidemiology, School of Public Health, Southern Medical University, Guangzhou 510515, China; xiongzhy7@163.com (Z.-Y.X.); hongminli2209@163.com (H.-M.L.); qiuchsh2021@gmail.com (C.-S.Q.); tangxulianpm@163.com (X.-L.T.); liaodq0926@163.com (D.-Q.L.); duliying02@163.com (L.-Y.D.); laishumin0119@163.com (S.-M.L.); huanghongxuan0113@163.com (H.-X.H.); zhangbingyun00@163.com (B.-Y.Z.); kuangling22@163.com (L.K.)

**Keywords:** gut microbiota, dementia, mendelian randomization

## Abstract

**Background**: The causal association of specific gut microbiota with dementia remains incompletely understood. We aimed to access the causal relationships in which one or more gut microbiota account for dementia. **Method**: Using data from the MiBioGen and FinnGen consortia, we employed multiple Mendelian randomization (MR) approaches including two-sample MR (TSMR), multivariable MR (MVMR), and Bayesian model averaging MR to comprehensively evaluate the causal associations between 119 genera and dementia, and to prioritize the predominant bacterium. **Result**: We identified 21 genera that had causal effects on dementia and suggested *Barnesiella* (OR = 0.827, 95%CI = 0.722–0.948, marginal inclusion probability [MIP] = 0.464; model-averaged causal estimate [MACE] = −0.068) and *Allisonella* (OR = 0.770, 95%CI = 0.693–0.855, MIP = 0.898, MACE = −0.204) as the predominant genera for AD and all-cause dementia. **Conclusions**: These findings confirm the causal relationships between specific gut microbiota and dementia, highlighting the necessity of multiple MR approaches in gut microbiota analysis, and provides promising genera as potential novel biomarkers for dementia risk.

## 1. Introduction

Dementia refers to a variety of neurodegenerative disorders characterized by the progressive deterioration of memory and cognitive functions [1]. The common types of dementia encompass Alzheimer’s disease (AD), vascular dementia (VaD), and frontotemporal dementia (FTD) [2]. In 2019, approximately 47 million individuals were living with dementia globally, and this number is projected to exceed 131 million by 2050 [3]. Given the lack of effective treatments for dementia, it is crucial to identify modifiable risk factors to reduce the economic and social impact of the disease [4].

The gut microbiota profoundly influences various aspects of host physiology such as hepatic metabolism [5], resistance to infection [6], and immune system development [7]. Emerging evidence suggests the existence of a bidirectional communication pathway between the gut microbiota and the brain, known as the microbiota gut–brain axis, which has potential for the prevention and treatment of dementia [8,9]. Observational studies have demonstrated that individuals with dementia exhibit significant differences in gut microbiota composition compared to healthy controls [10,11,12]. However, the robustness and causal implications of these findings frequently remain ambiguous. For instance, a cohort study (*n* = 77) reported a significant decrease in *Ruminococcaceae* and *Actinobacteria* in AD patients [13], whereas another case–control study (*n* = 86) found the opposite for these microbiota [14]. These discrepancies may be due to potential confounders and differences in sample characteristics, which limit the causal inference between gut microbiota and dementia risk.

Mendelian randomization (MR) is an approach that can overcome these limitations. By capitalizing on the stochastic inheritance of genotypes from one generation to the next, MR establishes a plausible causal pathway by ensuring that the relationship between genetic variations and outcomes remains unaltered by confounders [15]. Previous MR studies have evaluated the link between gut microbiota and dementia by two-sample MR (TSMR) [16,17], which leverages genetic variants associated with exposure and outcome from two separate samples, but lacks the consideration of other potential influences. For instance, Ji et al. [18] identified 36 gut microbiota that were causally associated with dementia through TSMR without considering the effects of multiple bacteria exposures. Given the potential interactions within the gut microbiota [19] that may bias the TSMR findings, conducting multivariate MR (MVMR) alongside TSMR is recommended. MVMR enables the examination of the direct effects of one exposure while controlling for others [20]. Furthermore, Bayesian model averaging MR (MR-BMA) enables ranking the gut microbiota in relation to dementia risk based on posterior probabilities (PP) [21], thereby prioritizing the predominant bacteria for dementia.

In this study, we aimed to apply multiple methods including TSMR, MVMR, and MR-BMA to comprehensively evaluate the causal effects of gut microbiota genus on dementia and prioritize the predominant causal gut microbiota for dementia, AD, VaD, and FTD.

## 2. Methods and Materials

### 2.1. Study Design

The design and overall workflow of this study are outlined in Figure 1B. After extracting IVs from GWAS, the TSMR analysis was initially performed to investigate the causal relationship between gut microbiota and dementia. Then, the gut microbiota that were statistically significant (*p* < 0.05) were selected for reverse MR analysis to examine the reverse causal links. Subsequently, we conducted MVMR and MR-BMA analyses to evaluate the impact of multiple exposures on one outcome and to identify the predominant gut microbiota for dementia, AD, VaD, and FTD.

### 2.2. Data Sources

Summary statistics for gut microbiota were obtained from a comprehensive multiethnic GWAS meta-analysis comprising 18,340 individuals across 24 cohorts, performed by MiBioGen [22]. They analyzed the microbial composition by 16S rRNA gene sequencing and identified a total of 211 taxa (131 genera, 35 families, 20 orders, 16 classes, and 9 phyla) [22]. We subsequently removed the unknown taxa, leaving a total of 196 gut microbiota constituents, with 119 genera used as exposure factors in this study.

Regarding the summary statistics for dementia, we obtained data from the FinnGen consortium R9 release, which provides comprehensive information on the association of genetic variations with multiple diseases. We extracted the following datasets: dementia (16,499 patients, 356,660 control individuals), AD (5422 patients, 356,660 control individuals), VaD (2335 patients, 360,778 control individuals), and FTD (111 patients, 360,134 control individuals). In addition, four datasets related to dementia, AD, VaD, and FTD were added as validation groups. FinnGen is a government business collaboration research initiative that integrates imputed genotype data with digital health record data from the nationwide longitudinal health register. This register has been collecting data from Finnish residents aged 18 years and older since 1969 [23]. Shi et al. reported that in situations where large sample sizes are used for summary statistics, populations are similar, and when the statistical analysis of exposure and outcome are derived from distinct coalitions, there is typically minimal overlap between samples of exposures and outcomes [24]. The data we utilized in our study met these criteria. Appendix A provides further details on the contributing GWAS consortia.

### 2.3. Instrumental Variables (IVs)

The MR approach relies on three core assumptions (Figure 1A): relevance assumption, independence assumption, and restriction-exclusion assumption [25]. The relevance assumption states that genetic variation should be strongly related to the risk factor of interest. The independence assumption implies that genetic variation should not be correlated with any confounders, while the restriction-exclusion assumption emphasizes that genetic variation should exclusively and causally pertain to the outcome through the studied exposure, without involving any alternative pathways [25]. To ensure the validity of our conclusions, we implemented various quality control measures to choose IVs. First, potential IVs were selected for gut microbiota based on single nucleotide polymorphisms (SNPs) that reached the locus-wide significance threshold (*p* < 1.0 × 10^−5^) [26]. Second, we utilized European sample data from the 1000 Genomes Project as our reference panel to assess the linkage disequilibrium (LD) between SNPs [27]. SNPs with an R^2^ value less than 0.001 and a clumping window size of 10,000 kb were selected to ensure variable independence. Third, we excluded SNPs with a minor allele frequency (MAF) ≤ 0.01 to mitigate bias introduced by rare variants and ensure the representativeness of our results [28]. Next, we determined the forward stranded alleles based on the allele frequency at which palindromic SNPs were present. Finally, we assessed the strength of instrumental variables (IVs) based on the *F*-statistic, which was calculated as follows: *F* = (R^2^ × (N − 1 − K))/(1 − R^2^) × K), where R^2^ represents the portion of exposure variance explained by the IVs, N represents the sample size, and K represents the number of IVs [29]. We excluded SNPs with *F*-statistics < 10 to avoid potential weak instrument bias.

### 2.4. Statistical Analyses

We applied the TSMR to estimate the effect of the single exposure of gut microbiota on dementia, AD, VaD, and FTD. Four statistical methods were used, comprising inverse-variance weighting (IVW) as the primary method, along with MR-Egger regression, weighted median, and MR pleiotropy residual sum and outlier test (MR-PRESSO). Assuming that each genetic variation meets the core assumptions, the IVW method is considered advantageous due to its ability to provide effect estimates that are robust to horizontal pleiotropy [30]. The MR-Egger regression method, by employing the independent assumption and examining the intercept term, allows for the assessment of pleiotropy [31]. The weighted median method complements MR-Egger regression by providing a valid estimate when at least 50% of the instrumental variables (IVs) are valid [32]. The MR-PRESSO method also requires at least 50% of the genetic variations to be valid. It is commonly employed to detect and mitigate horizontal pleiotropy by eliminating significant outliers [33]. In the event of discordance among the outcomes obtained from these approaches, we will accord with the results derived from the IVW approach as the primary outcome. We estimated the causal effects by odds ratios (ORs) and 95% confidence intervals (CIs), which represented an increased risk of dementia per SD increase in the abundance of gut microbiota features.

The gut microbiota identified by TSMR were included in our subsequent MVMR analysis, a statistical method for the estimation of causal effects using genetic variants associated with multiple risk factors [20], to estimate the causal effects on dementia in the context of multiple gut microbiota exposures. Furthermore, we employed MR-BMA to identify the gut microbiota that was predominant [21]. The MR-BMA, an extension of multivariable MR, combines multiple gut taxa using weighted linear regression models and evaluates the posterior probability (PP) of causality for each specific model within a Bayesian framework [21]. We calculated the marginal inclusion probability (MIP) by summing up the PP across all the models that incorporated the risk factor. Furthermore, we computed the model-averaged causal estimate (MACE) by averaging across these models, providing a conservative estimation of the direct causal impact of the risk factor on the outcome [21].

### 2.5. Sensitivity Analyses

To ensure the authenticity and robustness of our results, sensitivity analyses were conducted. Initially, we utilized the MR-Egger intercept test and MR-PRESSO global test to assess the existence of horizontal pleiotropy in the IVs (*p* > 0.05 indicated the absence of horizontal pleiotropy) [33]. Subsequently, we assessed heterogeneity using the Cochran *Q* statistic and Rucker *Q* statistic, with no heterogeneity indicated when *p* > 0.05 [33]. Additionally, we performed reverse MR analysis to exclude and evaluate the bi-directional causation effects between gut microbiota and dementia. Eventually, the screened bacteria will be incorporated into the validation set to assess the reliability of the results.

### 2.6. Software

We conducted all of the statistical analyses in R (version 4.2.1), employing the “TwoSampleMR” (version 0.5.6), “MendelianRandomization” (version 0.9.0), and “MR-PRESSO” (version 1.0) packages for MR analyses. For MR-BMA, we utilized the R code available on GitHub (https://github.com/verena-zuber/demo_AMD), accessed on 20 November 2023.

## 3. Results

### 3.1. Strength of the IVs

Associations between 119 gut microbiota and dementia including dementia, AD, VaD, and FTD were assessed. We employed 1526 SNPs across 119 bacterial genera following the criteria of IV selection, with a median of 12 SNPs for each genus. In addition, the *F*-statistics for each SNP ranged from 14.58 to 88.42, suggesting the absence of weak instrument bias (Appendix A). The change in IVs is visualized in Figure 1.

### 3.2. Two-Sample MR

The results of the TSMR revealed a significant association (*p* < 0.05, IVW method) between 21 gut microbiota and dementia (Table 1 and Appendix A). Specifically, we identified six genera associated with dementia, five genera associated with AD, five genera associated with VaD, and three genera associated with FTD (Figure 2 and Table 1).

### 3.3. MVMR and MR-BMA

The results of the MVMR and MR-BMA analyses are summarized in Figure 3 and Table 2. According to the MVMR results, we identified nine genera that had significant effects (*p* < 0.05) on dementia (one for dementia, five for AD, two for VaD, and two for FTD). *Barnesiella* (OR = 0.827, 95%CI = 0.722–0.948) demonstrated a potentially protective effect for dementia. For AD, *Allisonella* (OR = 0.770, 95%CI = 0.693–0.855), *Oscillibacter* (OR = 0.754, 95%CI = 0.633–0.898), and *Lachnospiraceae FCS020* (OR = 0.723, 95%CI = 0.566–0.924) showed protective effects, while *Intestinimonas* (OR = 1.462, 95%CI = 1.213–1.761) and *Defluviitaleaceae* UCG011 (OR = 1.292, 95%CI = 1.054–1.583) showed risk effects. For VaD, the two genera *Adlercreutzia* (OR = 0.589, 95%CI = 0.476–0.729) and *Veillonella* (OR = 0.682, 95%CI = 0.531–0.876) both demonstrated protective effects. For FTD, *Rikenellaceae RC9* (OR = 0.570, 95%CI = 0.328–0.990) and *Phascolarctobacterium* (OR = 0.156, 95%CI = 0.038–0.414) both evidenced protective effects. In the MR-BMA analysis, we found that four genera including *Barnesiella* (MIP = 0.464, MACE = −0.068), *Allisonella* (MIP = 0.994, MACE = −0.492), *Adlercreutzia* (MIP = 0.994, MACE = −0.492), and *Rikenellaceae RC9* (MIP = 0.533, MACE = 0.328–0.99) were the top-ranked genera correlated with dementia, AD, VaD, and FTD, respectively.

### 3.4. Sensitivity Analyses Results

The results of the sensitivity analyses encompassed the Cochran *Q* test, Rucker *Q* test, MR-Egger intercept test, and MR-PRESSO global outlier test (Table 1). There was no evidence of either heterogeneity or horizontal pleiotropy (all *p* > 0.05, Table 1) in the MR findings. Additionally, the IVs of dementia for reverse MR were extracted and are listed in Appendix A. As a result, except for the association between VaD and *Veillonella* (OR = 1.082, 95%CI = 1.014–1.154), no other bacteria demonstrated a reverse causal association with dementia (Appendix A). However, only two of the four predominant genera identified by MRBMA have successfully undergone external validation including *Barnesiella* (OR = 0.868, 95%CI = 0.781–0.964) and *Allisonella* (OR = 0.897, 95%CI = 0.815–0.987) (Appendix A).

## 4. Discussion

In this study, we conducted a comprehensive MR analysis to evaluate the causal relationship between gut microbiota and dementia, utilizing the summary statistics from MiBioGen and FinnGen. Our findings identified 21 causal associations between gut microbiota and dementia, while the MVMR analysis addressed 9 genera that exhibited causal effects on dementia in the context of multiple exposures. Furthermore, MR-BMA indicated *Allisonella*, *Adlercreutzia*, *Rikenellaceae RC9*, and *Barnesiella* as the predominant microbiota for dementia, with external validation further confirming the causal effects of *Allisonella* and *Barnesiella* on dementia.

Previous MR studies have identified several gut microbiota in relation to dementia risk [17,18,34,35], but few studies have considered the adjusted effects of multiple bacteria, which may lead to biased results. For example, we found that the elevated abundance of *Ruminococcus gnavus* was linked to decreased dementia risk, which is consistent with prior MR studies [18,34]. However, after adjusting for other genera associated with dementia risk in the MVMR analyses, the effect of *Ruminococcus gnavus* on dementia was no longer statistically significant (*p* > 0.05). Therefore, when multiple bacteria were identified, it was necessary to integrate TSMR and MVMR to validate these findings. Furthermore, the MR-BMA method demonstrated the capacity to discern the predominant bacteria across a range of gut microbiota. A previous MR study indicated that *Allisonella* and *Lachnospiraceae FCS020* were associated with an increased risk of AD [18]. However, it remained unresolved as to which bacterium was the predominant contributor to AD risk. Our findings support the association of *Allisonella* and *Lachnospiraceae FCS020* with increased AD risk while demonstrating the predominance of *Allisonella* among the gut microbiota associated with AD risk by MR-BMA. Consequently, the integration of TSMR and MR-BMA has the potential to assist in further selecting promising biomarkers for dementia.

Regarding the two genera, *Allisonella* and *Barnesiella*, achieved through the external validation process, the causal direction between *Allisonella* and cognition has yielded conflicting results in previous studies [34,36]. A cross-sectional study (*n* = 118) revealed that *Allisonella* was more abundant in individuals with mild cognitive impairment (MCI) than in the normal control group, indicating its potential risk to cognition [37]. Conversely, another study reported that *Allisonella* might be beneficial to cognition due to its potential role to protect the blood–brain barrier [36]. In our study, we demonstrated the protective effects of *Allisonella* on AD risk by comprehensive MR analysis. Although the mechanism between *Allisonella* and AD risk remains unknown, a reasonable hypothesis is that the association is mediated by histidine. *Allisonella* utilizes histidine as its sole energy source, and histidine plays a crucial role in regulating nerve inflammatory responses [38]. With regard to *Barnesiella,* it has been found to be associated with cognition in previous cross-sectional studies [39,40]. Jena et al. noted that *Barnesiella* is linked to the dysregulation of inflammatory factors as well as cognitive decline [39]. A decreased abundance of *Barnesiella* was also found to be associated with a greater cerebral small vessel disease burden [40]. Our MR findings favor the protective effect of *Barnesiella* on cognition and suggest a predominant role for *Barnesiella* in the gut microbiota implicated in dementia. Furthermore, the functional modules of *Barnesiella* were predicted to be associated with phenylpyruvate [40], which has previously been demonstrated to contribute to the progression of dementia [41]. Therefore, it is hypothesized that phenylpyruvate may be the mediator between *Barnesiella* and dementia risk, but further research is needed to clarify this mechanism.

In the case of the VaD and FTD groups, although the gut microbiota associated with these conditions were identified during the initial screening process, they did not pass external validation. It is hypothesized that the low number of cases sampled for VaD and FTD may have affected the statistical efficacy. Accordingly, with regard to the predominant bacteria suggested by MR-BMA, *Adlercreutzia* and *Rikenellaceae*, we consider this to remain a notable finding. Previous population studies have revealed that the abundance of *Adlercreutzia* bacteria is diminished in the AD group relative to healthy controls [42]. Conversely, *Rikenellaceae* has been observed to confer beneficial effects on cognitive function in the elderly [43]. This is the first study to report the association of *Adlercreutzia* and *Rikenellaceae* with VaD and FTD, therefore, further validation of this association with larger sample sizes is needed.

Our study has several strengths. First, MR analysis is less susceptible to reverse causation and confounding than traditional observational study designs. Second, we employed diverse methodological approaches including the TSMR, MVMR, and MR-BMA to comprehensively screen the layers of gut microbiota potentially associated with dementia. We also acknowledge the limitations of this study. First, the gut microbiota examined in this study was limited by the genus level of classification. This restricted the level of classification, which may not offer sufficient precision in detecting potential causal relationships. Although the genus level is currently the most detailed taxonomic unit in MiBioGen, a more detailed classification of the gut microbiota remains our future endeavor. Second, the genetic variations predominantly represent lifetime exposures rather than short-term interventions. Consequently, their ability to accurately capture the magnitude of short-term changes in the gut microbiota may be limited. Therefore, further clinical validation is required to ascertain the potential utility of the MR-screened gut microbiota as biomarkers. Third, although we implemented multiple sensitivity analyses to enhance the reliability of our causal estimates, the biological mechanisms underlying these genetic tools remain incompletely understood, presenting a challenge in definitively ruling out violations of the independence and exclusion limitation assumptions.

## 5. Conclusions

In this study, we comprehensively explored the causal associations between gut microbiota and dementia. We present evidence for *Allisonella* and *Barnesiella* as the predominant causal microbiota for dementia, highlighting the necessity of MVMR and MR-BMA as key additions to traditional MR analysis. By integrating multiple MR approaches, more robustness findings can therefore be obtained, which holds greater promise in biomarker identification for the early detection of dementia.

## Figures and Tables

**Figure 1 nutrients-16-03312-f001:**
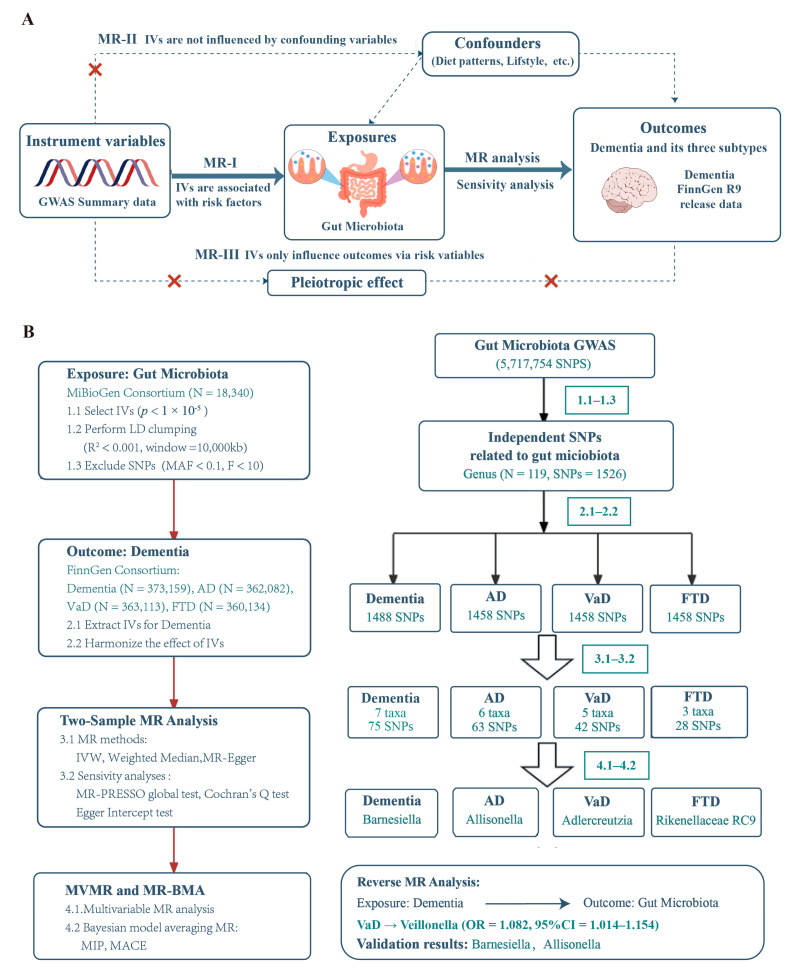
The study design of the MR analysis (**A**) and the overall workflow (**B**). GWAS: genome-wide association study; MR: Mendelian randomization; IVs: instrument variables; LD: linkage disequilibrium; SNP: single nucleotide polymorphism; IVW: inverse-variance-weighted; MR-PRESSO: MR pleiotropy residual sum and outlier; AD: Alzheimer’s disease; VaD: vascular dementia; FTD: frontal-temporal dementia.

**Figure 2 nutrients-16-03312-f002:**
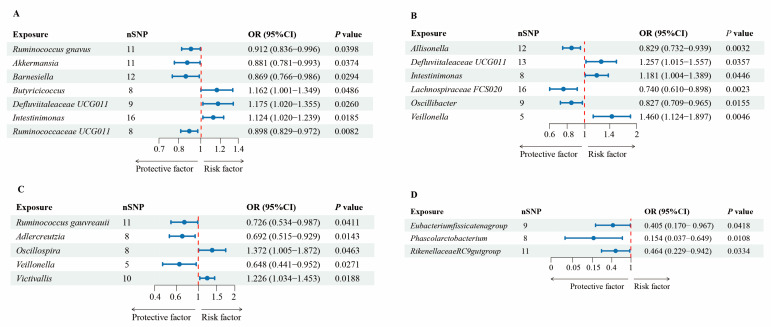
Forest plots of estimates identified with inverse-variance weighted. (**A**) Dementia, (**B**) Alzheimer’s disease, (**C**) vascular dementia, and (**D**) frontal-temporal dementia. The odds ratios (95% confidence interval) for dementia and its three subtypes per relative abundance increase in gut microbiota as estimated in the inverse-variance weighted MR analysis.

**Figure 3 nutrients-16-03312-f003:**
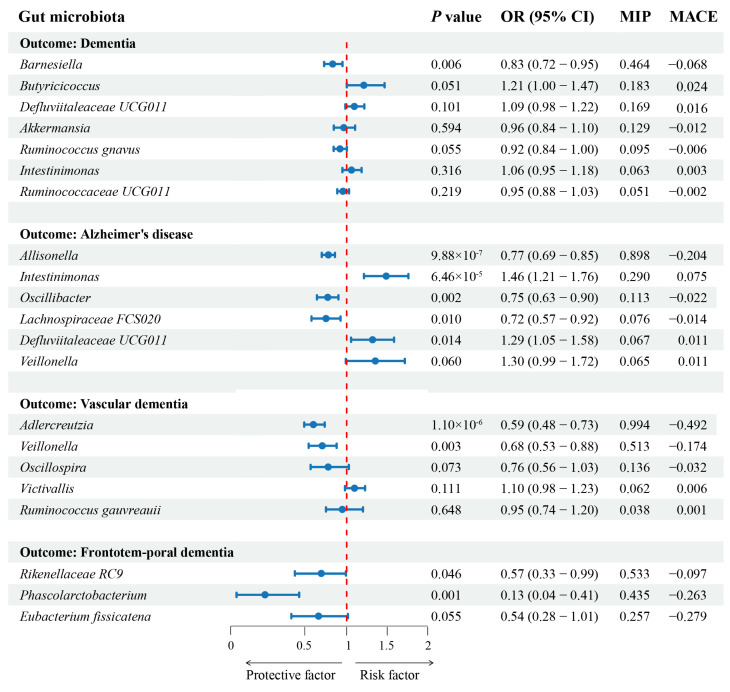
Multivariable MR and MR-BMA results for dementia and its three subtypes. The odds ratios (95% confidence interval) for dementia and its three subtypes per relative abundance increase in gut microbiota as estimated in the inverse-variance weighted MR analysis. The last column illustrates the MIP and MACE (MR-BMA output) to prioritize risk factors for dementia. MIP, marginal inclusion probability; MACE, model-averaged causal estimate.

**Table 1 nutrients-16-03312-t001:** MR results of significant relationships between gut microbiota and dementia.

Exposure	Outcome	Method	OR (95%CI)	*Q*	Intercept	Outlier
				Test	*p* Value	*p* Value
*Ruminococcus* *gnavus*	Dementia	IVW	0.912 (0.836–0.996)	0.435		0.511
		MR-Egger	0.779 (0.507–1.198)	0.393	0.481	
		WM	0.876 (0.778–0.987)			
*Akkermansia*	Dementia	IVW	0.881 (0.781–0.993)	0.395		0.468
		MR-Egger	0.987 (0.650–1.499)	0.336	0.589	
		WM	0.891 (0.757–1.048)			
*Barnesiella*	Dementia	IVW	0.869 (0.766–0.986)	0.836		0.790
		MR-Egger	1.028 (0.613–1.724)	0.806	0.527	
		WM	0.892 (0.760–1.048)			
*Butyricicoccus*	Dementia	IVW	1.162 (1.001–1.349)	0.423		0.486
		MR-Egger	1.420 (1.060–1.904)	0.591	0.170	
		WM	1.137 (0.922–1.402)			
*Defluviitaleaceae* *UCG011*	Dementia	IVW	1.175 (1.020–1.355)	0.147		0.193
		MR-Egger	0.852 (0.522–1.393)	0.209	0.224	
		WM	1.239 (1.061–1.446)			
*Intestinimonas*	Dementia	IVW	1.124 (1.020–1.239)	0.722		0.771
		MR-Egger	1.411 (1.076–1.849)	0.871	0.100	
		WM	1.086 (0.950–1.242)			
*Ruminococcaceae* *UCG011*	Dementia	IVW	0.898 (0.829–0.972)	0.338		0.418
		MR-Egger	1.037 (0.685–1.571)	0.290	0.513	
		WM	0.901 (0.818–0.993)			
*Allisonella*	AD	IVW	0.829 (0.732–0.939)	0.351		0.382
		MR-Egger	0.864 (0.347–2.155)	0.254	0.930	
		WM	0.853 (0.720–1.010)			
*Defluviitaleaceae* *UCG011*	AD	IVW	1.257 (1.015–1.557)	0.276		0.353
		MR-Egger	0.940 (0.424–2.084)	0.051	0.480	
		WM	1.310 (1.013–1.693)			
*Intestinimonas*	AD	IVW	1.181 (1.004–1.389)	0.740		0.712
		MR-Egger	1.316 (0.837–2.069)	0.254	0.622	
		WM	1.158 (0.922–1.455)			
*Lachnospiraceae* *FCS020*	AD	IVW	0.740 (0.610–0.898)	0.508		0.507
		MR-Egger	0.774 (0.459–1.304)	0.421	0.859	
		WM	0.767 (0.591–0.995)			
*Oscillibacter*	AD	IVW	0.827 (0.709–0.965)	0.979		0.981
		MR-Egger	1.007 (0.559–1.814)	0.976	0.511	
		WM	0.801 (0.654–0.980)			
*Veillonella*	AD	IVW	1.460 (1.124–1.897)	0.560		0.673
		MR-Egger	0.984 (0.087–11.064)	0.477	0.533	
		WM	1.412 (0.996–2.002)			
*Ruminococcus* *gauvreauii*	VaD	IVW	0.726 (0.534–0.987)	0.402		0.364
		MR-Egger	0.376 (0.108–1.311)	0.411	0.315	
		WM	0.864 (0.561–1.331)			
*Adlercreutzia*	VaD	IVW	0.692 (0.515–0.929)	0.430		0.503
		MR-Egger	0.719 (0.173–2.989)	0.322	0.959	
		WM	0.614 (0.417–0.902)			
*Oscillospira*	VaD	IVW	1.372 (1.005–1.872)	0.431		0.444
		MR-Egger	1.240 (0.302–5.093)	0.325	0.890	
		WM	1.345 (0.876–2.065)			
*Veillonella*	VaD	IVW	0.648 (0.441–0.952)	0.808	0.878	0.774
		MR-Egger	0.984 (0.445–21.928)			
		WM	0.763 (0.462–1.258)			
*Victivallis*	VaD	IVW	1.226 (1.034–1.453)	0.843		0.822
		MR-Egger	1.436 (0.391–5.281)	0.775	0.816	
		WM	1.218 (0.966–1.538)			
*Eubacterium* *fissicate*	FTD	IVW	0.405 (0.170–0.967)	0.911		0.910
		MR-Egger	0.265 (0.003–24.273)	0.856	0.857	
		WM	0.430 (0.144–1.288)			
*Phascolarcto* *bacterium*	FTD	IVW	0.154 (0.037–0.649)	0.436		0.487
		MR-Egger	0.044 (0.001–4.161)	0.367	0.563	
		WM	0.227 (0.033–1.584)			
*Rikenellaceae* *RC9*	FTD	IVW	0.464 (0.229–0.942)	0.994		0.991
		MR-Egger	0.112 (0.001–9.299)	0.994	0.539	
		WM	0.512 (0.206–1.269)			

The odds ratios (95% confidence interval) for dementia per relative abundance increase in gut microbiota. Abbreviations: MR: Mendelian randomization; AD: Alzheimer’s disease; VaD: vascular dementia; FTD: frontal-temporal dementia; CI: confidence interval; IVW: inverse-variance weighted; WM: weighted median; nSNP: number of single nucleotide polymorphisms; *Q* test: obtained *p* value from the Cochran *Q* test and Rucker *Q* test; intercept *P* value obtained from the MR-Egger intercept test; outlier *p* value obtained from the MR-PRESSO global outlier test.

**Table 2 nutrients-16-03312-t002:** Prioritization of predominant bacteria for dementia, AD, VaD, and FTD.

Gut Microbiota	Rank	MIP	MACE	PP
Dementia				
*Barnesiella*	1	0.46	−0.07	0.36
*Butyricicoccus*	2	0.18	0.02	0.12
*Defluviitaleaceae UCG011*	3	0.17	0.02	0.12
*Akkermansia*	4	0.13	−0.01	0.1
*Ruminococcus gnavus*	5	0.1	−0.01	0.07
*Intestinimonas*	6	0.06	0	0.05
*Ruminococcaceae UCG011*	7	0.05	0	0.04
AD				
*Allisonella*	1	0.9	−0.2	0.53
*Intestinimonas*	2	0.29	0.08	0.04
*Oscillibacter*	3	0.11	−0.02	0.01
*Lachnospiraceae FCS020*	4	0.08	−0.01	0.01
*Defluviitaleaceae UCG011*	5	0.07	0.01	0.01
*Veillonella*	6	0.07	0.01	0.01
VaD				
*Adlercreutzia*	1	0.99	−0.49	0.39
*Veillonella*	2	0.51	−0.17	0
*Oscillospira*	3	0.14	−0.03	0
*Victivallis*	4	0.06	0.01	0
*Ruminococcus gauvreauii*	5	0.04	0	0
FTD				
*Rikenellaceae RC9*	1	0.53	−0.1	0.37
*Phascolarctobacterium*	2	0.44	−0.26	0.27
*Eubacterium fissicatena*	3	0.26	−0.28	0.15

Abbreviations: AD: Alzheimer’s disease; VaD: vascular dementia; FTD: frontal-temporal dementia; MIP: marginal inclusion probability; MACE: model-averaged causal effect; PP: posterior probability.

## Data Availability

The data used in the present study are publicly available at the following websites: https://mibiogen.gcc.rug.nl/menu/main/home/ (accessed on 20 August 2023), https://www.finngen.fi/en/access_results (accessed on 20 September 2023 and 20 September 2024), and https://gwas.mrcieu.ac.uk/ (accessed on 20 September 2024).

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
