# Peer review of "Investigating Causal Associations between the Gut Microbiota and Dementia: A Mendelian Randomization Study"

_nutrients, 2024, doi:10.3390/nu16193312_

Round 1

Reviewer 1 Report

Comments and Suggestions for Authors

This article is a study on the causal relationship between gut microbes and dementia, some of the comments are as follows:

1.     Why this topic was chosen, there are many articles that have addressed the causal relationship between Gut Microbiota and Dementia. The main purpose of the article was not clearly defined and novelty was lacking. These need to be added in the introduction and abstract sections, in addition, the article needs to write in comparison with current references and list relevant points.

2.     Figure 1,2, and 3 is very poorly illustrated and needs to be sharpened.

3.     What is the difference between the authors' research and these two pieces of literature https://doi.org/10.1186/s12974-023-02999-0, https://doi.org/10.3389/fmicb.2023.1306048.

4.     What is the reason for the high p-value in Table 1, which needs to be further explained and illustrated in comparison with other articles

5.     Sensitivity analyses, reverse MR analysis results and Validation group MR analysis results need to be added.

6.     The conclusion of the article needs to be rewritten and the conclusion is unclear.

Comments on the Quality of English Language

Moderate editing of English language required.

Reviewer 2 Report

Comments and Suggestions for Authors

The manuscript of Xiong et al. Investigate causal associations between the gut microbiota and dementia. The paper  is presented already highlighted in yellow, even this the first round of revision (maybe an error during the submission?).

However, the manuscript is very good, here only few comments to improve the quality of paper.

Introduction

Line 27: "Dementia is a comprehensive classification...". Clarify whether dementia is considered a syndrome or condition. Use more standard terminology such as "Dementia refers to a range of neurodegenerative disorders characterized by..."

Results 

Line 170: "...revealing a significant association between 21 gut microbiota and dementia." "Significant" needs to be qualified with the level of significance (p-value thresholds, etc.). Clarify if this refers to statistically or biologically significant associations.

Discussion

Line 224: "Our results indicated the positive associations between 21 genetically predicted genera and dementia." Elaborate on why these findings are novel and how they contribute to the existing literature.

Line 265: "Our study has several strengths...".  Expand on the strengths, particularly the novelty of combining multiple MR approaches (TSMR, MVMR, MR-BMA). How do these approaches complement one another?

Conclusion

Line 282-283: "We conducted MR analysis to determine the causal association between 119 gut microbiota genera and dementia..." The conclusion can benefit from a stronger statement about the future directions of this research. Mention the potential for these findings to guide clinical interventions.

Comments on the Quality of English Language

Minor editing of English language required.

Round 2

Reviewer 1 Report

Comments and Suggestions for Authors

-